# Prevalence of Feeding Problems in Children and Associated Factors—A Cross-Sectional Study among Polish Children Aged 2–7 Years

**DOI:** 10.3390/nu15143185

**Published:** 2023-07-18

**Authors:** Agnieszka Białek-Dratwa, Oskar Kowalski

**Affiliations:** 1Department of Human Nutrition, Department of Dietetics, Faculty of Public Health in Bytom, Medical University of Silesia in Katowice, ul. Jordana 19, 41-808 Zabrze, Poland; okowalski@sum.edu.pl; 2Department of Cardiology, Congenital Heart Diseases and Electrotherapy, Silesian Center for Heart Diseases, 41-800 Zabrze, Poland

**Keywords:** neophobia, feeding difficulties, children, complementary feeding, baby-led weaning, BLW

## Abstract

Food neophobia is an aversion to eating or a reluctance to try unfamiliar or new foods. From an evolutionary perspective, this behaviour may minimise the risk of consuming foods that are harmful to health. However, such aversion causes food monotony, which may result in nutritional deficiencies. This study aimed to assess the prevalence of feeding problems among Polish children aged 2–7 years using the Montreal Children’s Hospital Feeding Scale and to investigate the correlation between age, gender, mode of feeding in infancy, including complementary feeding, and the prevalence of feeding difficulties in the study group of children. Material and method: The study group consisted of 585 children: 299 boys (51.11%) and 286 girls (48.89%). The study was conducted using a questionnaire-based method, with an indirect survey technique using a web-based form (CAWI). The research tool used was the Montreal Children’s Hospital-Pediatric Feeding Program. Results: Groups with the lowest risk feeding problems, risk 0, comprised 445 children (76.06%); group 1, middle difficulties, 59 children (10.08%); group 2, moderate difficulties, 40 children (6.84%); and group 3, most difficulties, 40 children (7.01%). The mean MCH-FS score for the entire study group was calculated and was 37.29 points ± 12.02; for 2 year olds, 35.69 points; for 3 year olds, 37.41 points; for 4 year olds, 38.31 points; for 5 year olds, 38.46 points; for 6 year olds, 37.95 points; and for 7 year olds, 36.06 points. The mean value of the MCH-FS scale for girls was 37.44 points, and for boys, 37.32 points. None of the above parameters correlated with the risk of feeding problems, including age, except with a non-significative tendency to be higher in the youngest age. Conclusion: Breast milk feeding and the time of complementary feeding (CF) in the study group did not influence the risk of feeding problems. Using the full BLW method during CF can protect the child against the occurrence of feeding problems such a food selectivity or picky eating in the future. In our study, children with difficulties during CF, mainly the vomiting reflex, were more likely to develop feeding problems such as food neophobia. Based on our study, we did not observe a correlation between age, gender, and the occurrence of feeding problems, and there was only a non-significant tendency to be higher in the youngest age. However, further research needs to be undertaken to assess how such behaviour affects subsequent feeding difficulties.

## 1. Introduction

Food neophobia is an aversion to eating or a reluctance to try unfamiliar or new foods [1,2]. From an evolutionary perspective, this behaviour may minimise the risk of consuming foods that are harmful to health. However, such aversion causes food monotony, which may result in nutritional deficiencies [1,3]. 

One theory of the prevalence of food neophobia is that in early human history, it was beneficial because it helped children avoid potential food hazards, including poisonous foods such as berries or spoiled meat. The ability to detect and avoid unfamiliar foods ensured that children only consumed safe, familiar foods they could trust. This innate preference for familiar foods was significant in the evolutionary pathway because it benefited the species’ survival [1,4]. 

However, the exact aetiology of food neophobia still needs to be determined, and its expression varies with age [5]. During the first period, infants consume one food: breast milk or infant formula. After 4 months of age, solid foods are gradually introduced into the infant’s diet, which may provide an opportunity for aversion to unfamiliar foods [6,7]. However, infants under 18–20 months of age readily accept new foods and do not show neophobic behaviour until 20–24 months [5,8]. Food neophobia can be observed in all age groups, but it increases rapidly during the complementary feeding period and peaks at 2–6 years of age [9].

In addition to evolutionary factors, environmental factors may also influence food neophobia in children. Parents and caregivers may inadvertently reinforce a child’s reluctance to try new foods by offering only familiar foods or expressing negative attitudes towards unfamiliar foods. This can lead to a vicious circle in which children become increasingly reluctant to try new foods, which can further limit the variety of foods they eat [10,11,12,13].

In addition, taste and texture preferences may play a role in food neophobia. Children may be more sensitive to bitter tastes or prefer sweet foods [14,15,16], making it difficult to savour new or unfamiliar foods. Therefore, it is important to introduce different flavours into the child’s diet to shape future food preferences [17]. Some studies suggest that breastfed children accept new foods more readily and have lower levels of food neophobia [18,19]. 

Babies fed with modified milk tend to become accustomed to the constant and specific taste of milk formula, consequently showing less tolerance or even aversion to new foods and tastes [15,16,20,21,22]. Studies have also shown that the more authoritarian the parents are during mealtimes, the more often the child rejects the foods offered [23,24,25,26].

Food neophobia is a complex phenomenon influenced by many factors, including evolutionary and environmental factors, taste preferences, and cultural norms. Understanding these factors can help parents and carers encourage children to try new foods and develop a more varied and balanced diet. Most studies of food neophobia have examined the prevalence of neophobia in different age groups rather than each year. This has led to broad peak estimates, and whether food neophobia differs between children aged 2 to 7 years remains unclear. Therefore, this study aimed to assess the prevalence of feeding problems among Polish children aged 2–7 years using the Montreal Children’s Hospital Feeding Scale and to investigate the correlation between age, gender, mode of feeding in infancy, including complementary feeding, and the prevalence of feeding difficulties in the study group of children. 

## 2. Materials and Methods

### 2.1. Course of the Study

The study was conducted using a questionnaire-based method, with an indirect survey technique using a web-based form (CAWI). The questionnaire was made available to mothers of children in randomly selected nurseries and kindergartens in Poland through closed groups on instant messaging systems designed for communication between parents and educational institutions and on parent association groups in individual cities and regions in Poland. All survey participants were informed about the purpose of the study, the voluntary nature of their participation, and the preservation of their anonymity, and were asked to accept the data-sharing rules. Adults (mothers of preschool and nursery-aged children) took part in the study. The study period covered the months of January to March 2023.

### 2.2. Selection of the Study Group

In verifying the study group of parents, it was observed that only one father completed the survey questionnaire; mothers completed the other questionnaires. Therefore, only mothers qualified for the study of feeding children during the period of dietary expansion and feeding during the nursery and preschool period, as they are the ones who are most often at home with their children during the complementary feeding period or in contact with an educational institution such as a nursery/preschool and are responsible for feeding children during this period. The survey was conducted using the CAWI method, so the sample selection was utterly random (according to the adopted inclusion and exclusion criteria of the survey).

### 2.3. Inclusion and Exclusion Criteria

Mothers taking part in the study gave their informed consent to participate, and the questionnaire was only made available when approval to participate in this study was obtained. The criteria for group selection included the fact that respondents were of legal age, had at least one child of nursery or preschool age, and had no formal knowledge of the behavioural determinants of nutrition (education or profession related to the topics of nutrition, treatment, and upbringing of children and adolescents). Inclusion criteria for the study proper were: being the mother/legal guardian of a child aged between 2 and 7 years, consent to participate in the study, and correct and complete completion of the questionnaire. The criteria for exclusion from the study were: lack of consent to participate in the study; incorrectly completed questionnaire, including non-response to questions; child’s age below 2 years and above 7 years; and the presence of a disease determining the method of feeding, e.g., food allergies and intolerances, autism spectrum disorders. After consideration of the inclusion and exclusion criteria, 585 pairs of mothers and their children were included in the final analysis.

The study was conducted according to the Declaration of Helsinki and the Act on the Profession of Physicians and Dentists. A positive opinion of the Bioethics Committee operating at the Medical University of Silesia in Katowice was obtained to conduct the research “Dietary neophobia among infants and children” (BNW/NWN/0052/KB/34/23).

### 2.4. Research Tool

The research tool was an anonymous survey questionnaire consisting of 5 parts. The first part concerned the parent/guardian and their child; it included data such as age and sex of the parent/guardian, place of residence, education of the parent/guardian, sex of the child, information on delivery, current weight and height, and food intolerances and allergies. Based on the child’s current age, weight, length/height using centile grids, and 3 SD BMI for girls and boys aged 0–3 years, the WHO standard body weight of children in terms of underweight, average weight, overweight, and obesity was assessed; for children aged 3–7 years, the developmental norms for girls/boys aged 3–18 years according to OLAF and OLAF studies [27,28] were used. The study asked for information entered in the “Child Health Booklet” such as the week of pregnancy in which the child was born, birth weight, birth length, and mode of delivery (natural, planned caesarean, unplanned caesarean) [29].

According to the Polish law issued by the Polish Ministry of Health, the child health booklet contains standardised information on the child, including the prenatal period; birth; health status after birth; patronage visits; preventive examinations, including dental examinations; history of infectious diseases, allergies, and anaphylactic reactions; radiological procedures; provision of medical devices; exemption from sports activities; and other information relevant to the assessment of the child’s normal development from birth, including measurements of weight, length/growth up to adult. All entries in the above document are made by medical staff, including a doctor, midwife, nurse, or other medical professional. This information is entered into the health booklet after providing the health service. If this is not possible, it is completed at the next visit based on the individual’s internal records [29].

The next part of the questionnaire focused on the mode of feeding during infancy, taking into account breast milk feeding, exclusive breastfeeding, length of breastfeeding, and the timing and method of CF (when the introduction of CF started; consistency of meals during CF—puree, pureed meals, meals ready to be eaten by the child with fingers; products given to the child as CF; and the method of expanding the infant’s diet including the use of the BLW method). In our study, we used the application of the baby-led weaning (BLW) method during complementary feeding. According to the definition of the BLW method, we considered that the child ate completely or mostly independently, so they ate using the BLW method; or the child was occasionally spoon-fed by an adult—approximately 10% feeding, 90% on their own, and also ate using the BLW method.

The third part concerned the child’s current diet, including the use of cutlery, food preferences, taste senses, feeding behaviour, and occurrence of food selectivity. The questionnaire was developed based on current dietary recommendations for the group of the youngest children and the method of diet expansion developed by PTGHiŻD [6] based on ESPHGAN recommendations [7], as well as information on diet expansion, including the BLW method and food selectivity occurring in this period of a child’s life [4,6,7,10,30,31].

The last part of the questionnaire concerned the prevalence of feeding problems. The research tool used was The Montreal Children’s Hospital-Pediatric Feeding Program [32,33]. The Polish version of the Montreal Children’s Hospital Feeding Scale (MCH-FS), which was appropriately translated and validated [33], was used in our study.

The Montreal Children’s Hospital-Pediatric Feeding Program (MCH-FS) is related to feeding a child from 6 months (receiving a pureed diet) to 6 years of age. It includes questions such as: How would you rate your child’s meal pattern? How concerned are you about your child’s meal pattern? How do you assess your child’s appetite (feeling of hunger)? At what point during a meal does your child start refusing to eat? How long do your child’s meals last (in minutes)? How do you assess your child’s behaviour during mealtimes? Does your child choke, gag, spit, or vomit at certain foods? Does your child hold food in the mouth without swallowing? Do you have to walk behind your child or distract him/her (toys, TV) to get him/her to eat? Do you have to force your child to eat or drink? How do you assess your child’s chewing (or sucking) skills? How do you assess your child’s growth (weight, height)? How does feeding your child affect your relationship with your child? How does feeding your child affect your family relationships? [32,33]. 

The MCH-FS consists of 14 items covering the following feeding characteristics: oral motility, sensory, and appetite. Other items address mothers’ concerns about feeding, mealtime behaviour, strategies mothers use, and the family’s reaction to the child’s feeding [32]. A 7-point Likert scale was included for each question. The meaning of the answers to the questions varied depending on the question [32,33]. Seven items from the MCH-FS scale are scored from a negative to a positive direction, and the remaining seven from a positive to negative direction (reversed scores). The final MCH-FS scale score is obtained by adding the scores for each question and reversing the scores of the seven items from negative to positive. The MCH-FS scale required an appropriate recalculation of the selected responses before adding up each respondent’s answers. Questions 1, 3, 4, 8, 10, 12, and 13 had to be reversed in order so that answer 1 had 7 points, answer 2 had 6 points, answer 3 had 5 points, etc. The remaining questions had to be kept in their original order; then, all the points obtained added together. The interpretation of the results was based on the study “The Montreal Children’s Hospital Feeding Scale: A brief bilingual screening tool for identifying feeding problems” by Maria Ramsay et al. from 2011 and the study “The Polish version of the Montreal Children’s Hospital Feeding Scale (MCH-FS): translation, cross-cultural adaptation, and validation)” by Katarzyna Bąbik et al. from 2019. The interpretation (MCH-FS) is that a score in the range of 14–45 points indicates no feeding difficulties; 46–52 points, middle difficulties; 53–58 points, moderate difficulties; above 59 points, most difficulties. We used the RAVE SCORE MCH-FS in the study, where the maximum score was 98 points [27,32,33].

### 2.5. Statistical Analysis

The programs used to analyse the collected data were Microsoft Office Word and Microsoft Office Excel. Statistical analysis was performed using Statistica v. 13.3 software (StatSoft Inc., Tulsa, OK, USA). The measured data were characterised by mean and standard deviation (X ± SD), and the range of minimum and maximum values obtained in the study group of children was determined. Statistical tests were used to analyse the variables for statistical inference. The study group of children was divided into 4 subgroups based on the calculation of the MCH-FS scale score: group 0, no risk; group 1, moderate difficulties; group 2, moderate difficulties; and group 3, most difficulties. Bivariate tables were used to compare the group of children in the four groups: 0, no risk; 1, middle difficulties; 2, moderate difficulties; 3, most difficulties, for r non-parametric characteristics Pearson’s chi-squared test was used. 

The level of statistical significance adopted in the study was set at *p* ≤ 0.05. 

## 3. Results

Characteristics of the Study Group

The study group consisted of 585 children: 299 boys (51.11%) and 286 girls (48.89%). Most of the children in the study came from cities with more than 100,000 inhabitants, i.e., 212 children (36.24). In terms of body weight, 404 children (69.06%) were of average weight, 139 children (23.76%) were underweight, 31 children (5.30%) were overweight, and 11 (1.88%) were obese. The study included children aged between 2 and 7 years of age, including 130 two year olds (22.22%), 134 three year olds (22.91%), 91 four year olds (15.56%), 80 five year olds (13.68%), 82 six year olds (14.02%), and 68 seven year olds (11.62%) (Table 1). Throughout the study, the grouping variable was allocated to the group at risk of feeding problems. Groups with the lowest risk of feeding problems (risk 0) comprised 445 children (76.06%); group 1, middle difficulties, 59 children (10.08%); group 2, moderate difficulties, 40 children (6.84%); and group 3, most difficulties, 40 children (7.01%). The correlation of feeding problem risk with gender (*p* = 0.988), place of residence (*p* = 0.755), weight (*p* = 0.755), and age (*p* = 0.764) was analysed. None of the above parameters correlated with the risk of feeding problems, including age, except for a non-significative tendency to be higher in the youngest age.

However, it can be seen in the results that the prevalence of feeding problems is lowest in 2 year olds (84.61%), and the level then increases in children aged 7 years.

The mean MCH-FS score for the entire study group was calculated and was 37.29 points ± 12.02; for 2 year olds, it was 35.69 points; for 3 year olds, 37.41 points; for 4 year olds, 38.31 points; for 5 year olds, 38.46 points; for 6 year olds, 37.95 points; and for 7 year olds, 36.06 points. The mean value of the MCH-FS scale for girls was 37.44 points, and for boys, 37.32 points. Age and gender differences were not statistically significant. 

By definition, “exclusive breastfeeding” means not giving modified milk to the baby; the baby consumes only breast milk. Both the length of breast milk feeding (*p* = 0.242), the length of exclusive breast milk feeding (*p* = 0.296), and the time of initiation of complementary feeding (*p* = 0.899) did not correlate with the risk of feeding problems.

The mode of complementary feeding in the children studied was also assessed. Children given puree during CF were more likely to have a higher risk of feeding problems (*p* = 0.010). The administration of puree with lumps did not correlate with the risk of feeding problems (*p* = 0.240). The study also included the baby-led weaning (BLW) method, which involves child-controlled feeding. It is based on the omission of the spoon-feeding stage by the parents/carers and the feeding of pulpy foods (purees). When the baby can sit up unaided (approx. 6–7 months of age), various solid foods are given in such a form that they can be easily grasped with the hand (e.g., cucumber strips, carrots, pieces of apple, pear, various shapes of pasta, and strips of meat) [6,7]. Children in whom the BLW method was used in dietary expansion did not show a risk of feeding problems (*p* = 0.026), in contrast to children fed traditionally with a spoon. The study also assessed the difficulties in introducing new foods into the child’s diet. Children who experienced difficulties during CF were more likely to be at risk of feeding problems (*p* < 0.001) (Table 2).

Table 3 shows the results for the occurrence of problems during CF. Children who had a vomiting reflex during CF were more likely to have feeding problems (*p* = 0.001). In contrast, spitting food out of the mouth (*p* = 0.085), gagging (*p* = 0.244), choking (*p* = 0.590), and choking and needing medical attention (*p* = 0.121) did not correlate with the risk of feeding problems. 

Table 4 and Table 5 show the mothers’ subjective assessment of their child’s food intake and appetite and of the fact that their child was a picky eater. In both cases, the majority of mothers correctly assessed their child’s appetite and the fact of being a picky eater (*p* < 0.001) and (*p* < 0.001).

Among the children surveyed, the consumption of foods with specific tastes was assessed. In the surveyed group, most children consume products with different flavours (90.43%). A total of 4.44% of the children surveyed consume products with selected flavours (Table 6).

The study assessed current problems related to children’s eating patterns. Both the vomiting reflex (*p* < 0.001, V Cramer = 0.274), spitting food out of the mouth (*p* < 0.001, V Cramer = 0.289), playing with food (*p* = 0.004, V Cramer = 0.149), and burying cutlery in food (*p* = 0.001, V Cramer = 0.186) correlated positively with the occurrence of feeding problem risk. Choking did not correlate with feeding problems (*p* = 0.278, V Cramer = 0.0810) (Table 7). 

Table 8 analyses the principal components of the Montreal Children’s Hospital Feeding Scale (MCH-FS) for the entire sample of children. The mean, along with the standard deviation obtained for each question, and the median, were assessed. The higher the mean and median, the more frequent the behaviour. In the study group, the most frequent behaviour occurred in the aspect of walking behind the child or distracting the child (toys, TV) in order for the child to eat a meal (mean 5.67 ± 1.80, median = 7). Frequent behaviours were as follows: the child refusing to eat a meal (mean 3.19 ± 1.91, median = 3), extending the meal time (mean 3.13 ± 1.41, median = 3), and forcing the child to eat and drink (mean 3.16 ± 2.48, median = 3).

## 4. Discussion

There is no precise formal definition of picky eating, although it is generally accepted that it includes the rejection or restriction of familiar and unfamiliar foods, and thus includes an element of neophobia; these factors are associated with feeding problems in children [9,31]. The most common definition offered by Dovey et al. is that picky/fussy eaters are children “who consume an inappropriate variety of foods, rejecting a significant amount of foods that are familiar (as well as unfamiliar) to them”. They see food neophobia (food aversion or avoidance of new foods) as a somewhat distinct construct, while observing that the two factors of food aversion and avoidance of new foods are interrelated and both contribute to food rejection or acceptance, especially of fruit and vegetables [9]. Hence, the terms feeding neophobia (used according to Dovey’s definition), feeding problems, and picky eating will appear frequently in our study and in this discussion. One would consider food neophobia to be a fear of new, unfamiliar foods, but many of the authors of the studies cited below consider children with food neophobia to be children who have an aversion to eating or who avoid new foods (according to Dovey’s definition). Food aversion and avoidance of new foods, in turn, lead to a variety of feeding problems such as refusal to eat particular foods because of their texture, taste, colour, or shape, reflexes such as the vomiting reflex, or spitting out food or closing the mouth.

A high prevalence of food neophobia and pickiness has been reported previously among children aged 3 to 7 years in a study by Hafsrad G. et al. [34]. An exceptionally high prevalence of picky eating and food neophobia has been reported previously in China (59%) and the United States (60%) [35]. In contrast, a study in the Netherlands showed a very low prevalence of picky eating (5.6%) among 4 year old children [36]. A Polish study by Kozieł-Kozakowska et al. conducted among children aged 2.5–7 years showed low neophobia in 12.3% and high neophobia in 10.8% of the children studied [14]. In the meta-analysis by Torres et al. [2], the prevalence of food neophobia was present in 10 (53%) of the studies analysed and ranged from 12.8% to 100%.

In diagnosing feeding difficulties, including pickiness and feeding neophobia, the Montreal Children’s Hospital Feeding Scale (MCH-FS) may be applicable [33,37]. The Children’s Feeding Related Scale (MCH-FS Scale) is used to screen children with feeding difficulties adding challenges for preventive and diagnostic purposes. It can be used as a perfect tool to identify possible feeding difficulties in children aged 6 months to 6 years. The MCH consists of 14 questions addressing issues related to the course of the meal, assessment of appetite evaluation, meal duration, problems within the orofacial sphere, or the parent’s perception of the child’s average weight and height. Each question is answered on a 7-point Likert-type scale [33]. The MCH-FS can be used in the nutritional interview and can be extended to include questions about the child’s diet.

Feeding problem, selective eating, and food neophobia can lead to deficiencies in some essential nutrients, especially vitamins and minerals [1]. Children with high levels of food neophobia and other different feeding problems showed reduced adherence to standard eating patterns, which can negatively affect dietary diversity and lead to imbalanced nutrient intake [38]. This is supported by studies by Yong [39], Schmidt [40], Bell [41], and Kaar [13] Falciglia [3].

The results of the Di Nucci study showed a low intake of foods typical of the Mediterranean model, such as fruit, vegetables, and legumes, and conversely, a high intake of foods typical of the Western dietary model, such as sweets, sugary drinks, and red meat [42].

Food neophobia tends to occur with highly recommended and health-promoting foods such as fruit, vegetables, and legumes, which taste bitter or sour. Lower intake also occurs in the group of animal products, such as fish [43]. Children with high food neophobia were more likely to consume ultra-processed, sugar-rich foods (snacks, filled and unfilled cakes and sweets), as well as protein-rich foods (white meat, cheese and yoghurt) [38].

Some studies indicate that neophobic children are less likely to meet the recommended intake standards, especially the need for vitamin E [3]. In addition, children who only eat selected foods may not acquire specific eating skills, especially if they only eat soft-textured or pureed foods [1]. The negative health consequences of food neophobia should be seen in the context of the lost potential health benefits of a poor or poorly varied diet and, above all, the consumption of too few vegetables and fruits compared to recommendations [3,44]. A review of publications shows that neophobic children have a deficient intake of fruit and vegetables, which are health-promoting. This is supported by most epidemiological studies that demonstrate the health-promoting effects of fruit and vegetable consumption [45].

Nutritional neophobia is highly relevant to the concept of metabolic programming, especially in the nutritional aspect, which is understood as the long-term or lifelong effect of a stimulus or signal affecting the structures or functions of the organism during a critical period of development. It has been shown that the occurrence of factors such as malnutrition, or nutrient deficiency or excess, during so-called critical periods can reprogram the metabolism, leading to irreversible consequences. The first 1000 days of a child’s life is when the metabolism is programmed, and many of the physiological processes responsible for appetite control and energy regulation are fixed. During the first three years of a child’s life, there is less activity of enzymes produced in the liver, which is responsible for the metabolism of harmful compounds. The immature kidneys do not yet excrete toxins efficiently. The child’s diet should include, above all, products rich in vitamins A, D, C, and B. Micronutrients such as zinc, selenium, iron, and copper should also be present. Polyunsaturated fatty acids, including DHA, cod liver oil, probiotics, and prebiotics, are also essential to the diet [46,47,48,49]. Therefore, it seems essential to minimise the occurrence of nutritional neophobia in the youngest children. The above study aimed to answer questions on how feeding neophobia can be minimised through early modification of feeding behaviour from birth through CF.

The origin of food neophobia can be traced back to evolution when a neophobic attitude protected children from eating potentially contaminated food [14]. Humans, as an omnivorous species, had to distinguish between safe and poisonous food to survive [50]. Although this skill has now lost its value, it can still be observed in children around and after the age of 2 years, when unfamiliar foods or foods given in a different way than before cause anxiety in the child and a relative preference for familiar foods is apparent [51].

Although food neophobia is genetically determined, environmental factors that underlie individual differences in taste preferences can also influence its occurrence [50]. Genetic factors influencing food choice are related to taste receptors, which can differentiate the perception of sweet, umami, or bitter tastes depending on individual gene differences [51]. Thus, some children tolerate bitter-tasting green vegetables such as broccoli or cabbage better, others will not care for them, and some will reject these foods at the mere sight of them [51]. In our study, regardless of the level of risk of feeding problems, the majority of the children (90.43%) consumed products from different taste groups, 4.44% of the children only consumed products with selected tastes, and only one child out of the entire group consumed products with a sweet or bitter taste.

Food preferences are highly variable, resulting in a reluctance to eat new foods, and those less accepted may be reduced in the child. This is influenced by several factors, including the diet during pregnancy and lactation [52] or the mode of exposure and its repetition [51]. These are essential factors that may indirectly influence feeding difficulties and the course of food neophobia, which, depending on individual characteristics, may go unnoticed.

In our study, the length of breast milk feeding (*p* = 0.242), the length of exclusive breast milk feeding (*p* = 0.296), and the time of initiation of complementary feeding (*p* = 0.899) did not correlate with the risk of feeding problems. The Øverby study found no significant association between feeding problems and any breastfeeding nor a significant correlation between exclusive breastfeeding [53]. Maier’s study compared the acceptance of new foods by milk-formula-fed and breastfed infants when they received different foods at different frequencies. They found that breastfeeding and milk formula feeding and giving a variety of foods during the early weaning period, rather than giving a specific food, often resulted in better acceptance of new foods, as measured several weeks after the intervention [18].

There are not many studies linking the method of dietary expansion and the occurrence of feeding problems. In our study, we verified the use of the BLW method by assessing the estimated percentage of spoon-feeding during CF. The children we included in the group of children who were fully fed using the BLW method were labelled as entirely or mostly independent eaters, and children who were occasionally spoon-fed by an adult (approximately 10% adult feeding 90% independent) (full-BLW). Children using the complete BLW method in CF did not show a risk of feeding problems (*p* = 0.026), unlike children fed traditionally with a spoon. It should be emphasised that we assessed the full use of the BLW method in the present analysis. The mixed BLW method (50% adult feeding and 50% self-feeding) already indicated a higher risk of feeding problems. Since our study was not aimed at verifying the BLW method as a superior method, we believe that the study should be extended in this respect to fully assess in which aspect the BLW method helps avoid the occurrence of feeding problems.

Other factors also influenced the course of feeding problems. In a study by An M. et al., the main factors were urging the child to eat with a firm refusal on the child’s part, unpleasant emotions during mealtimes (e.g., parent’s nervousness, stress, child’s crying), and high levels of neophobia in the mother [54]. Similar conclusions were reached by de Oliveira Torres et al. [2] in a systematic review of the literature, stating that the level of neophobia in children is influenced by, among other things, the eating habits of the parents, children’s innate preference for sweet and salty tastes, the mismatch between texture and the child’s psychomotor skills, pressure during meals, failure to read hunger and satiety signals, and monotony in child feeding. In our study, children who experienced a vomiting reflex during complementary feeding were at higher risk of developing food neophobia (*p* = 0.001), whereas spitting food out of the mouth (*p* = 0.085), gagging (*p* = 0.244), choking (*p* = 0.590), and choking and needing medical intervention (*p* = 0.121) did not directly affect the risk of developing feeding problems at a later stage of development.

The high prevalence of feeding problems between the ages of 2 and 6 years may also be because children tend to behave assertively and try to become independent from their parents. Therefore, refusing certain foods is a way of asserting their authority and presence. Another reason for this higher figure may be that older children are influenced by their peers and family, making them more likely to accept new foods [55,56].

The factors influencing food neophobia and feeding problems vary widely. On the one hand, it is a natural developmental stage; on the other hand, some factors may influence the perpetuation of inappropriate behaviour [54]. Therefore, appropriate intervention should be undertaken if neophobic behaviours do not subside but intensify. As in the case of eating disorders, the patient should be managed by a team of specialists, including a paediatrician/gastroenterologist, a clinical dietician, a neurologist, a psychologist, a sensory integration therapist, and a feeding therapist [2,37].

## 5. Conclusions

The following conclusions can be drawn from this study: Breast milk feeding and the time of CF in the study group did not influence the risk of feeding problems.Using the full BLW method during CF can protect the child against the occurrence of feeding problems, such a food selectivity or picky eating, in the future.In our study, children with difficulties during CF, mainly the vomiting reflex, were more likely to develop feeding problem such as food neophobia.In our study, we did not observe a correlation between age, gender, and the occurrence of feeding problems; there was only a non-significant tendency to be higher in the youngest age.However, further research needs to be undertaken to assess how such behaviour affects subsequent feeding difficulties.

## 6. Strengths and Limitations of the Study

The results of our study should be interpreted with its limitations in mind. The study should expect some risk of error due to the greater interest of study participants in their children’s diets.

Our study was a retrospective study, which may influence the occurrence of the false memory effect, especially in the group of mothers of older children, particularly 4–7 year olds, regarding the details of CF and infancy.

Additionally, the survey was conducted using the CAWI method, which is repeatedly criticised for lacking insight into the data collection process. However, it is worth noting that this type of data collection is widely accepted and convenient for collecting large amounts of information in groups that are often difficult to reach.

The MCH-FS tool was not developed primarily for screening and diagnosing neophobia; however, using the questions in this questionnaire, we believe that many of the questions and answers provided can be a tool for initial screening of food neophobia.

The advantage of the study is the size of the group; to date, most studies on food neophobia among children have been conducted on smaller groups of subjects. It is also worth mentioning that very few studies have been conducted on this topic, especially in Poland, and the above study is also currently being continued. In addition, to date, no cross-sectional study has examined the relationship between the use of the BLW method and difficulties during CF and the CF method, and the occurrence of food neophobia in the preschool age group.

## Figures and Tables

**Table 1 nutrients-15-03185-t001:** Characteristics of the study group of children with a breakdown according to the MCH-FS scale (0—no risk, 1—middle difficulties, 2—moderate difficulties, 3—most difficulties).

	0—Not at Risk		1—Middle Difficulties		2—Moderate Difficulties		3—Most Difficult		Total		
	n	%	n	%	n	%	n	%	N	%	
The entire group of children surveyed	445	76.06	59	10.08	40	6.84	41	7.01	585	100	
Gender:											*p* = 0.988
Boy	228	76.25	29	9.69	21	7.02	21	7.02	299	100	
Girl	217	75.87	30	10.48	19	6.64	20	6.69	286	100	
Body weight:											*p* = 0.755
underweight	98	70.50	16	11.51	13	9.35	12	9.63	139	23.76	
normal weight	314	77.72	36	8.91	25	6.18	29	7.17	404	69.06	
overweight	25	80.64	5	16.12	1	3.23	0	0.00	31	5.30	
obesity	8	72.72	2	18.18	1	9.09	0	0.00	11	1.88	
Age:											*p* = 0.764
2 years	110	84.61	12	9.23	4	3.07	4	3.07	130	22.22	
3 years	100	74.62	13	9.70	11	8.20	10	7.46	134	22.91	
4 years	66	72.52	9	9.89	8	8.79	8	8.79	91	15.56	
5 years	59	73.75	9	11.25	5	6.25	7	8.75	80	13.68	
6 years	62	75.06	7	8.53	6	7.31	7	8.53	82	14.02	
7 years	48	70.58	9	13.23	6	8.82	5	7.35	68	11.62	

**Table 2 nutrients-15-03185-t002:** Mode of complementary feeding in study children with respect to the MCH-FS scale and interpretation of the MCH-FS scale.

Method of Complementary Feeding	0—Not at Risk		1—Middle Difficulties		2—Moderate Difficulties		3—Most Difficult		Total		
	n	%	n	%	n	%	n	%	N	%	
puree	369	82.92	57	96.61	38	95.00	39	95.12	503	85.98	*p* = 0.010
puree with lumps	316	71.01	44	74.58	30	75.00	28	68.29	418	71.45	*p* = 0.240
Feeding/spoon fed:											*p* = 0.026
The child ate completely or											
mostly independently (BLW)	69	88.46	3	3.85	2	2.56	4	5.13	78	13.33	
Child occasionally spoon-fed by an adult (approximately 10% feeding, 90% on their own) (BLW)	11	100.00		0.00		0,00		0.00	11	1.88	
Baby fully or mostly spoon-fed by an adult	106	68.83	17	11.04	13	8.44	18	11.69	154	26.32	
The child was half fed by an adult with a spoon. half ate independently	259	75.73	39	11.40	25	7.31	19	5.56	342	58.46	
Difficulties in introducing new foods in the child: I don’t remember	19	57.58	6	18.18	3	9.09	5	15.15	33	5.64	*p* < 0.001
No, there were no problems in expanding the baby’s diet	349	84.50	36	8.72	16	3.87	12	2.91	413	70.60	
Yes, there were problems in expanding the child’s diet	77	55.40	17	12.23	21	15.11	24	17.27	139	23.76	

**Table 3 nutrients-15-03185-t003:** Incidence of problems during complementary feeding in the study group of children with respect to the MCH-FS scale and interpretation.

Problems during CF		0—Not at Risk	1—Middle Difficulties	2—Moderate Difficulties	3—Most Difficult	Total						
		n	%	n	%	n	%	n	%	N	%	
Vomiting reflex	yes	113	25.39	14	23.73	16	40.00	21	51.22	164	28.03	*p* = 0.001
spat food out of its mouth	yes	270	60.67	38	64.41	29	72.50	32	78.05	369	63.08	*p* = 0.085
Gagging	yes	142	31.91	12	20.34	14	35.00	15	36.59	183	31.28	*p* = 0.244
Choking	yes	33	7.42	4	6.78	4	10.00	1	2.44	42	7.18	*p* = 0.590
Choked and needed medical attention	yes	1	0.22	0	0.00	0	0.00	1	2.44	2	0.34	*p* = 0.121

**Table 4 nutrients-15-03185-t004:** Mothers’ subjective assessment towards her child’s food intake and appetite score (*p* < 0.001).

	0—Not at Risk		1—Middle Difficulties		2—Moderate Difficulties		3—Most Difficult		Total	
	n	%	n	%	n	%	n	%	N	%
The child often does not want to eat and I have to encourage/force him to do so	36	28.80	33	26.40	28	22.40	28	22.40	125	21.37
The child has an appetite and eats almost everything he is given.	297	98.67	3	1.00	0	0.00	1	0.33	301	51.45
The child has an appetite but eats a severely limited amount of food (up to 20 dishes)	8	61.54	1	7.69	3	23.08	1	7.69	13	2.22
The child has an appetite, but eats a limited amount of food	23	85.19	2	7.41	2	7.41	0	0.00	27	4.62
The child doesn’t want to eat, but I don’t force him to.	63	64.95	19	19.59	6	6.19	9	9.28	97	16.58
Child has no appetite, but eats because he is very hungry (eats only a limited amount of food)	1	50.00	0	0.00	0	0.00	1	50.00	2	0.34
Baby has no appetite, but eats because he is very hungry (eats everything)	0	0.00	0	0.00	1	100.00	0	0.00	1	0.17
I didn’t pay attention to it.	17	89.47	1	5.26		0.00	1	5.26	19	3.25

**Table 5 nutrients-15-03185-t005:** Mother’s subjective assessment of their child being a picky eater (*p* < 0.001).

The Fact of Being a Picky Eater	0—Not at Risk		1—Middle Difficulties		2—Moderate Difficulties		3—Most Difficult		Total	
	n	%	n	%	n	%	n	%	N	%
no	372	92.08	23	5.69	9	2.23		0.00	404	69.06
I don’t know	31	62.00	10	20.00	4	8.00	5	10.00	50	8.55
yes	42	32.06	26	19.85	27	20.61	36	27.48	131	22.39

**Table 6 nutrients-15-03185-t006:** Consumption of flavoured meals in the study group of children (*p* < 0.001).

	0—Not at Risk		1—Middle Difficulties		2—Moderate Difficulties		3—Most Difficulties		Final Total	
	n	%	n	%	n	%	n	%	n	%
I don’t know/difficult to say	13	46.43	5	17.86	6	21.43	4	14.29	28	4.79
No, he consumes products from different taste groups.	419	79.21	51	9.64	30	5.67	29	5.48	529	90.43
Yes, he eats only bitter-tasting products	0	0.00	0	0.00	1	100.00	0	0.00	1	0.17
Yes, he eats only sweet-tasting products	0	0.00	0	0.00	0	0.00	1	100.00	1	0.17
Yes, he only consumes products with the flavour of his choice.	13	50.00	3	11.54	3	11.54	7	26.92	26	4.44

**Table 7 nutrients-15-03185-t007:** Current problems related to the child’s eating patterns.

Problems Related to Eating		0—Not at Risk		1—Middle Difficulties		2—Moderate Difficulties		3—Most Difficult		Total		
		n	%	n	%	n	%	n	%	n	%	
vomiting reflex	yes	6	1.35	3	5.08	5	12.50	8	19.51	22	3.76	*p* < 0.001
spitting food out of mouth	yes	31	6.97	13	22.03	11	27.50	15	36.59	70	11.97	*p* < 0.001
playing with food	yes	131	29.44	23	38.98	17	42.50	22	53.66	193	32.99	*p* = 0.004
burying cutlery in food	yes	129	28.99	27	45.76	23	57.50	18	43.90	197	33.68	*p* = 0.001
whooping	yes	2	0.45	1	1.69	1	2.50	1	2.44	5	0.85	*p* = 0.278

**Table 8 nutrients-15-03185-t008:** Principal component analysis of the Montreal Children’s Hospital Feeding Scale (MCH-FS) for the whole children sample.

Factors and Items	Construct	Mean	Median
1. how do you find mealtimes with your child?	Parental concern	2.93 ± 1.52	3
2. how worried are you about your child’s eating?	Parental concern	2.67 ± 1.83	2
12. how do you find your child’s growth?	Parental concern	1.86 ± 1.43	1
3. how much appetite (hunger) does your child have?	Appetite	2.70 ± 1.61	2
4. when does your child start refusing to eat during mealtimes?	Appetite	3.19 ± 1.91	3
9. Do you have to follow your child around or use distractions (toys, TV) so that your child will eat?	Compensatory strategies	5.67 ± 1.80	7
10. Do you have to force your child to eat or drink?	Compensatory strategies	3.16 ± 2.48	1
6. how does your child behave during mealtimes?	Mealtime behaviour	2.79 ± 1.07	1
13. How does your child’s feeding influence your relationship with him/her?	Family relations	2.07 ± 1.47	1
14. How does your child’s feeding influence your family relationships?	Family relations	2.26 ± 1.72	1
5. how long do mealtimes take for your child (in minutes)?	Compensatory strategies	3.13 ± 1.41	3
7. does your child gag or spit or vomit with certain types of food?	Oral sensors	1.55 ± 1.07	1
11. how are your child’s chewing (or sucking) abilities?	Oral motor	1.46 ± 1.15	1
8. Does your child hold food in his/her mouth without swallowing it?	Oral sensory, oral motor, mealtime behaviour	1.83 ± 1.38	1

## Data Availability

The data presented in this study are available on request from the corresponding author. The data are not publicly available due to restrictions that apply to the availability of these data.

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
