# Peer review of "Prevalence of Feeding Problems in Children and Associated Factors—A Cross-Sectional Study among Polish Children Aged 2–7 Years"

_nutrients, 2023, doi:10.3390/nu15143185_

Round 1
Reviewer 1 Report
REVIEW
“Prevalence of feeding neophobia and feeding difficulties - a cross-sectional study among Polish children aged 2-7 years”
This study is transversal, descriptive, aiming to report the prevalence of neophobia and associated factors in 585 children from 2 to 7 years old. The authors applied the Montreal Children's Hospital - Pediatric Feeding Program Scale (MCH-FS) as the tool for measure it, but this tool was not developed for screening or diagnosing neophobia, having only one question for indirectly approximate to it.
Although it is a very interesting topic, it has to be rewritten, changing the title to “Prevalence of feeding problems in children and associated factors” and change the text and the references related to neophobia.
Detailed comments:
ABSTRACT
- Introduction: It must relate to and specify the objective of the study.
- Results: Mean test scores for MCH-FS are shown, but the prevalence of neophobias is not displayed here.
- Conclusions: any of all the conclusions emerge from the reported results.
INTRODUCTION
The introduction is well written.
What I missed is that neophobias should be recognized as a physiological process of maturation, and specify when they are pathological.
At the end of it, the authors line four objectives:
1. To assess the prevalence of feeding neophobia among Polish children aged 2-7 years using the Montreal Children's Hospital Feeding Scale
2. To investigate the correlation between age, gender, mode of feeding in infancy, including complementary feeding and the prevalence of feeding difficulties in the study group of children.
3. To verify the appropriateness of MCH-FS for children aged 7 years.
4. To describe the self-assessment of the mothers of the children towards the prevalence of feeding neophobia among their children.
I think that they must focus on objectives 1 and two. Number three needs a different type of analysis, and number four is not precise, because no direct question about neophobia was formulated.
METHODS
The research tool (MCH-FS) was conceived to easily detect the feeding problems in children, it doesn’t specify neophobias.
In this section the authors don’t define Neophobia as a variable and assimilate the results of the test to “risk of feeding neophobia”. They also change the scores originally defined for mild (61 to 65), moderate (65-70) and severe (>70) feeding difficulties, to numbers under those (46-52, 53-58 and above 59 points, respectively, without justification.
RESULTS
In this section, there are too many tables, with results that are not relevant, and could be only referred in the text.
Table one describes the prevalence of feeding difficulties (named this time “risk of neophobia)”, showing no differences in the comparisons according to sex and age. In lines 218-219 they describe that the prevalence is lowest in the 2 y.o. group, but in parenthesis they give the % of no risk of them. They also state that was higher in 7 y.o., but without a statistical difference.
After that they describe in tables 2, 3 and 4, the relationship with breastfeeding (length and exclusivity), time of complementary food initiation, and found no differences.
Related to the mode of feeding (table 5), there were differences between the risk groups related to give a pureed food, and to the only question indirectly related to neophobias (“difficulties to introduce new food in the child diets”).
In lines 252 to 262, they describe for first time the BLW method, that should be defined in the methods section, and that is not well-defined trough this question.
Next, problems during complementary feeding, vomiting reflex would be more frequent, but the size of this sub-sample is too low.
Tables 7 shows the mothers' subjective assessment of their child's food intake and appetite. Table 8 shows if their child is an eater or is vegan?
Table 9 shows the consumption of foods with specific tastes, most children consume products with different flavors.
Table 10 shows current problems related to the child's eating patterns.
Finally, table 11 analyses the principal components of MCH-FS scale for the entire sample,
DISCUSION
In the discussion different concepts are used as synonyms of neophobia, most of them correspond to food refusal and ARFID (selective type), and they are not a refusal to eat a new food for first time. Should be shorter and focused to the discussion without repeating the results.
In strengths and limitations it is well discussed the representativity of the sample, because of its selection, and the retrospective nature and false memory effect. But it is not included the limitation of the tool used (MCH-FS), that was not developed for screening or diagnosing neophobia, having only one question for indirectly approximate to it.
The conclusions are not supported by the results:
1. Prevalence by age was not significatively different.
2. -
3. BLW was not defined or asked properly in the scale.
4. Only follow up studies can support this conclusion.
Author Response
Dear Reviewer
Thank you very much for your time and all your valuable suggestions to make our manuscript more understandable to the reader.
Responding to your questions and suggestions for changes, we have decided to describe one by one what has been changed in our manuscript. At the same time, we have marked all the changes we made after the reviews of the two reviewers in red to make it easy to follow the changes.
Abstract
1 We have changed the title to Prevalence of feeding problems in children and associated factors as you suggested.
2 We have added the purpose of the study to the introduction. We have changed the word neophobia to feeding problems in the objective.
3 We have edited the conclusions
Introduction and objectives
We removed objective 3 and 4. And in objective 2 we changed the word feeding neophobia to feeding difficulties.
Methods
We added a rationale for our scoring, which was used in the Polish interpretation of the MCH-FS. We have used the RAVE SCORE score rather than the converted to 100 - the T-SCORE. We added a sentence regarding our scoring.
We used the RAVE SCORE MCH-FS in the study, where the maximum score was 98. We removed issues related to feeding neophobia, leaving only feeding difficulties.
Which in the following results in the elimination of terms related to neophobia.
Results
Table 1 We include results for differences between gender and age. But the differences are not statistically significant between the groups. However, referring to age, differences are observed (although not statistically significant) it is 84.61% of two-year-olds have a lack of risk of feeding problems, while among 7-year-olds this result is much higher 70.58%. Hence, we decided to describe this result.
In Table 2,3,4 there are no statistically significant differences, but we feel that these results are significant, i.e. that there is no correlation between the length of breastfeeding, exclusive breastfeeding, and the time of introduction of complementary feeding.
Table 5 - this is why we decided to change the title of the paper.
BLW - the use of the BLW method was not the main objective of our study, we added a description of BLW in the methodology.
Table 6 - please note that here is the number of only those children who had some problems while eating and "yes" answers were marked. In this table we did not describe those who answered "no" to the above question - we did not want to present results that are not relevant. For us, the "yes" answers were relevant. But the size of this group combining 'yes' and 'no' answers is 585 children - i.e. the entire study group answered this question.
In Table 8, the Polish term for a child who does not want to eat as 'vegan child' was mistranslated. We are very sorry for our error. The correct term would be "picky eater child".
In the limitations of the study, we described that the MCH-FS tool was not designed for screening and diagnosing neophobia, however, many of the questions that were used in the MCH-FS scale can be helpful in assessing food neophobia among children, e.g. how would you rate the course of a meal? At what point during a meal does the child start refusing to eat? How long do meals last? Does the child vomit during meals/choke? All these questions can also help to catch children who have feeding problems such as food neophobia.
The conclusions have been partly rewritten according to suggestions.
1 - We have explained our approach to this question above.
2. -
3. we have added a description of BLW in the methodology.
4. we have reworded this proposal.
Again, thank you very much for your valuable time and every suggestion to make us manuscript very good.
Best regards
Authors

Reviewer 2 Report
The work has been exhaustive and very complete. Its presentation is quite adequate, but there are points in which I think it should be improved:
- In lines 195-199 the definition of the groups is not completely clear and apparently repeated,
- The p values are normally given with only 3 decimal places, which makes it easier to read and understand.
The presentation of the tables is complicated, I think it would be substantially improved if the n(%) format were followed, with which the number of columns would decrease almost by half.
- A figure should be made with the variables to be highlighted, as said form of graphic presentation is more illustrative.
Author Response
Dear Reviewer,
Thank you very much for your time and all your valuable suggestions to make our manuscript more understandable to the reader.
Responding to your questions and suggestions for changes, we have decided to describe one by one what has been changed in our manuscript. At the same time, we have marked all the changes we made after the reviews of the two reviewers in red to make it easy to follow the changes.
In the description of the MCH-FS, which is a standardised questionnaire, we followed the classic description of the study. I.e.
The MCH-FS scale required an appropriate recalculation of the selected responses before adding up each respondent's answers. Questions 1, 3, 4, 8, 10, 12 and 13 had to be reversed in order so that answer 1 had 7 points, answer 2 had 6 points, answer 3 had 5 points and so on. The remaining questions had to be kept in their original order, then all the points obtained added together. The interpretation of the results was based on the study 'The Montreal Children's Hospital Feeding Scale: A brief bilingual screening tool for identifying feeding problems' by Maria Ramsay et al. from 2011 and the study 'The Polish version of the Montreal Children's Hospital Feeding Scale (MCH-FS): translation, cross- cultural adaptation, and validation) by Katarzyna BÄ…bik et al. from 2019.
The p-values have been revised and reported to 3 decimal places as suggested.
The tables are very large in terms of the number of results, the example given by the Reviewer will make the tables even less visible. We hope that in the final submission stage of the manuscript, the Editor will help us to format the tables in such a way that they are more readable by the reader. I am aware that there are a lot of results in our tables, but we cannot not present them all.
Please clarify if you are referring to a graphical abstract? Should we convert any of the tables you indicate into a figure. The graphical abstract has been sent to the Editor.
Again, thank you very much for your valuable time and every suggestion to make us manuscript very good.
Best regards
Authors

Round 2
Reviewer 1 Report
This is my 2nd review of the study, now renamed:
PREVALENCE OF FEEDING PROBLEMS IN CHILDREN AND ASSOCIATED FACTORS- A CROSS-SECTIONAL STUDY AMONG POLISH CHILDREN AGED 2-7 YEARS
The authors accepted many of my recommendations, but I think there are still aspects to improve to accept it for publication, that are now detailed:
ABSTRACT:
Results:
- Lines 25-28: The authors only report the means for the (MCH-FS) test at each age range, they don’t report any prevalence, that is the principal objective of the study.
- They should include here the global prevalence of feeding problems reported on lines 239 to 242, and pointed that there was NO relation to age, only a non-significative tendency to be higher in the youngest age.
Conclusions:
- The conclusions are not supported for any result in the previous section.
- They must be deleted, leaving only the global prevalence, if they want to leave the relation to vomit or BLW, these results must be written in the Results section of the abstract.
INTRODUCTION:
- Line 77: the right word is “authoritarian” for this phrase. From the reference n°27:
(Parents can score high or low on each of the dimensions, resulting in a four-fold classification of parenting styles: (1) authoritative (high demandingness/high responsiveness); (2) authoritarian (high demandingness/low responsiveness); (3) indulgent (low demandingness/ high responsiveness); and (4) uninvolved (low demandingness/ low responsiveness). Many studies on general parenting behaviours have found that authoritative parenting is associated with the most positive child outcomes (e.g., Darling & Steinberg, 1993; Maccoby & Martin, 1983). Similarly, in relation to feeding, the authoritative style of providing rules but in a positive context is associated with the development of the healthiest eating habits (Ventura & Birch, 2008), including greater fruit and vegetable intake (Blissett, 2011).
METHODS
- Lines 179-180: In this phrase the authors state “The last part of the questionnaire concerned the prevalence of feeding neophobia. The research tool used was The Montreal Children's Hospital - Pediatric Feeding Program” but they don’t define neophobia as a variable, and it is not specified which question of the (MCH-FS) refers to it.
RESULTS
- There are too many tables, I recommend:
o Table 1 simplify the size of inhabitants of the city.
o Tables 2 to 4 could be deleted, and let these results only cited in the text, because there are no signification in the differences.
o Tables 5, 6, 7, 8, 9 and 10: not to repeat the results in the text.
o Tables 8 and 9: simplify, leaving only the associations with significative difference. In table 9: 1st line, correct "3 dif bridge".
- Lines 279-282 refers to neophobia twice; it must be changed.
- Lines 307-310: This is a non-sense sentence (“Tables 7 and 8 show ………. which is confirmed by the results presented in Tables 7 and 8”).
DISCUSSION:
The discussion is still mostly referred to neophobias, and different concepts are used as synonyms of it, most of them correspond to food refusal and ARFID (selective type). I recommend changing for a more general concept as “feeding problems” because there are not necessarily neophobias (the refusal to eat a new food for first time). Could be still shorter and focused, without repeating the results.
The conclusions must be fixed:
1. Prevalence of neophobias was not studied. The difference of feeding problems by age was not significatively different.
2. ok
3. Not a projection, only the conclusion.
4. ok
Author Response
Dear Reviewer
Thank you very much for all your comments and we hope that after further revisions our publication will be even clearer.
We have the biggest problem with defining the term food neophobia. As we understand that the issue of food neophobia is related to a reluctance to eat new and unfamiliar foods, however, according to many authors, food neophobia is treated as picky eating, we have therefore added the following text to the discussion:
There is no precise formal definition of picky eating, although it is generally accepted that it includes the rejection or restriction of familiar and unfamiliar foods, and thus includes an element of neophobia, which are associated with feeding problems in children [62, 63]. The most popular definition proposed by Dovey et al. defines fussy children as "eating an inappropriate variety of foods, rejecting a significant amount of foods that are familiar (as well as unfamiliar) to them". They view food neophobia (food aversion or avoidance of new foods) as a somewhat distinct construct, while noting that the two factors of food aversion and avoidance of new foods are interrelated and both contribute to food rejection or acceptance, especially of fruit and vegetables [62]. Hence, the terms food neophobia (used according to Dovey's definition), feeding problems and food choosiness will appear frequently in our study and in this discussion. One might consider food neophobia to be a fear of new, unfamiliar foods, but many of the authors of the studies cited below consider children with food neophobia to be those with food aversion or avoidance of new foods (as defined by Dovey). Food aversion and avoidance of new foods, in turn, lead to various feeding problems, such as refusal to eat certain foods because of their texture, taste, colour, shape, reflexes such as the vomiting reflex, spitting out food or closing the mouth.
To put it all in perspective. For us, this phrase 'food aversion or avoidance of new foods' is important - which is why we wrote about eating problems as food neophobia in the article.
To make it easier for you to follow our changes, we have highlighted them in blue.
They relate to your comments:
Summary:
- We have added the prevalence of feeding problems based on MCH-FS.
- We added information that there was no association with age, only a non-significant tendency to be higher at the youngest age.
- Conclusions in the abstract: We have added those that relate to the results presented in the abstract.
Introduction:
- line 77 - we have changed to 'authoritarian' as recommended
Methods
- line 179-180 - we have changed to feeding problems
Results
- table 1 - we removed the breakdown by place of residence - it adds nothing to the study, these data are not used in any way
- tables 2,3,4 removed
- Tables 5,6,7,8,9,10 - we believe that some of the results should be described, but we have left those we consider necessary so that the reader knows what to pay attention to when reading the text
- Tables 8 and 9 - we feel that we cannot simplify them any further. We have corrected the "bridge"
- line 279-282 - corrected
- line 307-310 - deleted
Discussion
We have tried to plot as much as possible, but we feel we need to compare some of our results with the results of others, so we quote them. We have also added the explanation we wrote about above.
We have described feeding problems more generally than food neophobia, but many of the authors to whom we refer our results use the phrase food neophobia rather than feeding problems or food choosiness - we had to leave this out. I hope you can understand why we did this.
Conclusion
1) We have removed
2. remained
3. has been rewritten on request
4. remained
We have added one more conclusion
Based on our study, we did not observe a correlation between age, gender and incidence of feeding problems, there was only a non-significant trend towards being taller at the youngest age.
Once again, thank you very much for all the corrections and your time. We hope that we have improved everything in this round of reviews so that the publication is of the highest standard.
Greetings
Authors
